# Associations of intimate partner violence and financial adversity with familial homelessness in pregnant and postpartum women: A 7-year prospective study of the ALSPAC cohort

Caitlin S. Chan[1]*, Aaron L. Sarvet[1], Archana Basu[1], Karestan Koenen[1], Katherine M. Keyes[2]

1 Harvard T.H. Chan School of Public Health, Boston, MA, United States of America, 2 Columbia University Mailman School of Public Health, New York, NY, United States of America

* cac115@mail.harvard.edu

## Abstract

### Objective

To determine whether emotional and physical intimate partner violence (IPV) and financial adversity increase risk of incident homelessness in pregnancy and the post-partum period.

### Study design

Data were drawn from the Avon Longitudinal Study of Parents and Children, which starting in 1990 mailed questionnaires to 14,735 mothers in the UK, over 7 years from pregnancy onwards. Marginal structural models and multiple imputation were used to address time-varying confounding of the primary variables, testing for interaction between concurrent emotional/physical IPV and financial adversity, and adjusted for baseline age, ethnicity, education, partner's alcohol use, parity, depression, and social class.

### Results

Emotional IPV (HR 1.44 (1.13,1.84)), physical IPV (HR 2.05 (1.21,3.49)), and financial adversity (HR 1.59 (1.44,1.77)) each predicted a multiplicative increase in the discrete-time hazard of incident homelessness. We identified joint effects for concurrent emotional IPV and financial adversity (HR 2.09 (1.35,3.22)) and concurrent physical IPV and financial adversity (HR 2.79 (1.21,6.44)). We further identified a temporary decline in self-reported physical IPV among mothers during pregnancy and up to 8 months post-partum.

### Conclusions

Emotional and physical IPV and financial adversity independently and jointly increase the risk of incident homelessness. The effects of emotional and physical IPV are comparable to or greater than the risk of financial adversity. Homelessness prevention policies should consider IPV victims as high-risk, regardless of financial status. Furthermore, self-reported

**Data Availability Statement:** Access to ALSPAC research data must be requested using the formal procedures described in this document (http://

www.bristol.ac.uk/media-library/sites/alspac/documents/researchers/data-access/ALSPAC_Access_Policy.pdf) and is subject to eligibility, the ALSPAC funder's terms and conditions and University of Bristol policies and procedures. Data are available upon request from the ALSPAC executive committee (alspac-exec@bristol.ac.uk) for researchers who meet the criteria for access.

**Funding:** The UK Medical Research Council and Wellcome (Grant ref: 217065/Z/19/Z) and the University of Bristol provide core support for ALSPAC. This publication is the work of the authors and Caitlin Chan will serve as guarantor for the contents of this paper. A comprehensive list of grants funding is available on the ALSPAC website. This research was specifically funded by Wellcome Trust (Grant ref: WT092830/Z/10/Z Funder URL: https://wellcome.org/grant-funding) This study was supported in part by NIMH grants U01MH110925 (AB and KCK) and T32 MH 017119 (ALS) (Funder URL: https://www.nimh.nih.gov/index.shtml) The funders had no role in study design, data collection and analysis, decision to publish, or preparation of the manuscript.

**Competing interests:** The authors have no competing interests to declare.

physical IPV declines temporarily during pregnancy and up to 8 months post-partum. Screening for IPV in this period may miss high-risk individuals.

## Introduction

Violence remains highly prevalent in the private sphere of the home [1]. An estimated 15%-71% of women have experienced physical, emotional, or sexual violence from a romantic partner [2], collectively known as intimate partner violence (IPV). While IPV affects individuals of all genders, it is the primary cause of death by homicide among women, and is a risk factor for a wide range of poor mental, physical, socio-economic, and intergenerational outcomes [3–8], including homelessness. In fact, retrospective, qualitative, and case-control studies have identified IPV as a major risk factor for homelessness [9–11]. Homelessness remains a major public health problem. In 2017, 80,000 households in the UK were homeless, the highest number since 2007, and 45% of these households were single mothers with dependent children [12]. In the US, 1.3 million households with children in public schools experienced homelessness in the 2015–2016 school year [13]. Given that IPV can be economically destabilizing for women [14, 15], and that gender-based economic inequity may arise from the greater burden of parenting on women [16], the transition to parenthood is particularly relevant to examine.

In the US, domestic violence was the third most cited cause of familial homelessness [17]. This paper examines the contribution of IPV to risk of homelessness in a prospective cohort.

Two recent cohort studies with robust sample sizes provide heterogeneous estimates of the effect of IPV on subsequent housing instability. In 2018, Montgomery, Sorrentino, Cusack, et al [18] examined clinical screenings and medical claims in the Veterans Health Administration, documenting that those who had experienced IPV were 4 times as likely to screen positive for homelessness. In 2016, Dillon, Hussain, Kibele, Rahman, and Loxton [19] found a 1.14–1.21 increased odds of domestic relocation associated with recent IPV among a cohort of 5000 Australian women. However, both studies share limitations including reliance on complete case data. Given that women who experience IPV and homelessness are more likely to drop out of studies, such biases may attenuate the association between IPV and housing instability and homelessness.

While the association between IPV and homelessness is well-documented, the existing literature suffers from methodological limitations that make it challenging to assess whether IPV is a causal risk factor for homelessness. First, although studies have found an association between IPV and homelessness or housing instability, most data are cross-sectional [20] or use qualitative data from small sample sizes [21–23]. Thus, temporality is impossible to establish. Second, studies have not fully examined the role of financial adversity in the association between IPV and homelessness. Importantly, the ability to articulate temporal pathways in the relationship of IPV as a predictor of homelessness has been limited by the structure of available data. Given that women who have experienced IPV and become homeless are more likely to be in poverty, assessments of valid associations need to include identification strategies that untangle the effects of co-occurring financial adversity from the effects of IPV on homelessness. Third, as mentioned above, cohort studies have used complete case analysis, which is likely biased by underreporting of IPV and homelessness.

With these factors controlled, we hypothesize that, within a UK cohort of mothers, emotional IPV and physical IPV are independently drivers of incident homelessness, and that concurrent financial adversity modifies this risk.

## Methods

### Study design

This prospective, observational study estimates prevalence and life course trends of IPV on incident homelessness in a cohort of mothers over seven years.

### Setting and participants

The women in this study were the primary caregivers of children in the Avon Longitudinal Study of Parents and Children (ALSPAC) birth cohort, a prospective study based in the South West of England that has tracked children and their families across three generations. Ethical approval for the study was obtained from the ALSPAC Ethics and Law Committee and the Local Research Ethics Committees (Reference number C2481), and all participants provided written informed consent. ALSPAC Executive committee has been registered with/has an Institutional Review Board status with the United States Department of Health and Human Services and Office for Human Research Protections (IRB I00003312). The ALSPAC committee approval serves as the IRB approval. The full list of Research Ethics Committees who approved the study can be accessed online (http://www.bristol.ac.uk/media-library/sites/alspac/documents/governance/Research%20Ethics%20Committee%20approval%20references.pdf)

Most of the children were born in 1991–1992; additional children were later recruited and added to the cohort. Detailed information about the ALSPAC cohort is available in existing literature [24, 25]. The initial group consisted of 14,541 pregnancies. The mothers were residents of Avon while pregnant with expected delivery dates between April 1, 1991 and December 31, 1992. When the initial cohort of children were seven years old, further opportunistic recruitment of children who met the original eligibility criteria increased the cohort data. The cohort is more affluent and racially homogenous than the overall UK population. Over 99.7% of the women identified as heterosexual when their children were 7 years old, at the close of our study period [26].

From the ALSPAC cohort of 15,445 births, we excluded twin births. Of the remaining 15,039 records, we excluded two records with the same study ID. Finally, we excluded records where children were missing all study variables, leaving 14,735 observations. These data were used to create an imputed dataset for our primary analysis.

### Variables

Time-varying variables were measured via mailed self-reported questionnaires completed by the study mothers about themselves and about the primary child across multiple time points. The questionnaires are available on the study website (http://www.bristol.ac.uk/alspac/researchers/our-data/questionnaires/). The data used are from eight time points based on the child's age, from pregnancy at eighteen weeks gestation past 6 years of age of the study child. S1 Table in the S1 Appendix shows each wave of data collection (in relation to child's age) and the prior interval of time each assessment's responses referred to, which ranged from 18 weeks (at wave 1) to 17 months (at wave 6). Please note that the study website contains details of all the data available through a fully searchable data dictionary and variable search tool (http://www.bristol.ac.uk/alspac/researchers/our-data/).

**Exposure variables.**   Two items were used to identify IPV. At each of 8 waves, physical IPV and emotional IPV by the partner to the mother were measured as binary variables (Y/N). Physical IPV was indicated by the mother's response to the statement "Your partner hurt you physically [within the previous interval of time]." Emotional IPV was indicated by the mother's

response to the statement "Your partner was emotionally cruel to you [within the previous interval of time]." Both physical IPV and emotional IPV were defined in previous studies [27].

Exposure to financial adversity was measured at each wave by the mother's response (Y/N) to the statement "You had a major financial problem [in the prior interval of time]."

To model the multiplicative increase in the discrete-time hazard of incident homelessness for each additional count of emotional IPV, physical IPV, and financial adversity, we created three cumulative measures defined as the total count over time of affirmative responses to each exposure item at each interval. Thus, cumulative physical IPV, emotional IPV, and financial adversity scores had a maximum value of 1 in wave 1 and a maximum value of 8 in wave 8.

**Outcome variable.** Homelessness was measured at each wave of data collection by the dichotomous response (Y/N) to the statement "You became homeless [within the previous interval of time]." The study outcome was time to first incidence of homelessness.

**Baseline covariates.** The following sociodemographic covariates, measured at baseline or as close to baseline as available, were identified as possible confounders or effect measure modifiers.

Mother's age was measured in years. Maternal ethnicity was categorized as white or non-white. Mother's highest education qualification at baseline was categorized as CSE (Certificate of Secondary Education) or less, vocational, O level, A level and degree. The UK Registrar General's occupational coding "social class" variable was used to indicate mother's socio-economic status.

Maternal parity was measured at baseline. Postpartum depression severity was measured with the Edinburgh Postnatal Depression Scale (EPDS) [28] score.

The earliest measure of partner alcohol use available (reported by the mother at child age 8 months/wave 3 and indicating whether the partner had "alcoholism" since the child was born), was used as a measure of baseline partner alcohol use disorder.

## Missing data

During the study period, there was increasing loss-to-follow-up of study participants at each wave of data collection. While the overall cohort participation declined with each wave, the pattern of missingness was not monotonic—some participants missed one or several surveys between waves in which their data were collected. Table 1 displays the numbers of missing observations at first and last waves for the four main variables in our analysis: physical IPV, emotional IPV, financial adversity, and homelessness. Trends in missing data over the course of follow-up were roughly equivalent across main exposure variables (IPV) and financial adversity, ranging from around 10% at wave 1 to 43% at wave 8. To mitigate potential bias from missing data, we performed multiple imputation by chained equations (MICE) using fully conditional specification (FCS) for our primary analysis.

MICE was performed in SAS with the proc mi command, with 70 burn-in iterations. Among our cases, 57% were missing data for at least one variable contributing information to our full model, so we ran 57 imputations [29]. Logistic regression with augmented likelihood was used to impute dichotomous variables. Categorical variables were estimated with the discriminant function, and linear regression was used to estimate continuous variables.

Predictor variables for imputation included all variables used in the study model. Additional predictors were chosen for their possible association with differential inclusion and drop-out rates of exposure and outcome groups. Details on predictor variables are in the technical appendix (List A in the S1 Appendix).

A sensitivity analysis was conducted to address concerns of model misspecification when imputing missingness using FCS. We performed MICE using the non-parametric random forest

**Table 1. Characteristics of study population (First and last wave measurements are listed for time-varying variables).**

| N = 14,735 | N Missing | Original cohort | Data imputed with MICE using FCS |
|---|---|---|---|
| | | Prevalence (%) or Mean (Std Dev) | Prevalence (%) or Mean (Std Dev) |
| *Primary study variables* | | | |
| Any Physical IPV | 1483 | 9.4% | 20.2% |
| Physical IPV: Wave 1 | 3034 | 1.7% | 4.0% |
| Physical IPV: Wave 8 | 6389 | 2.6% | 7.5% |
| Any Emotional IPV | 1490 | 24.4% | 36.6% |
| Emotional IPV: Wave 1 | 3071 | 6.1% | 9.9% |
| Emotional IPV: Wave 8 | 6391 | 8.0% | 14.4% |
| Any major financial problem | 1629 | 35.7% | 54.8% |
| Major financial problem: Wave 1 | 3021 | 13.7% | 31.4% |
| Major financial problem: Wave 8 | 6372 | 9.8% | 14.9% |
| Any homelessness | 1479 | 4.8% | 13.1% |
| Homeless: Wave 1 | 3023 | 1.4% | 3.2% |
| Homeless: Wave 8 | 6370 | 0.6% | 2.1% |
| *Socio-demographic variables* | | | |
| Age at delivery | 1029 | 28.0 (5.0) | 27.7 (4.6) |
| Ethnic background | 2661 | | |
| White | | 97.4% | 93.4% |
| Non-white | | 2.6% | 6.6% |
| Highest education qualification | 3323 | | |
| CSE or less | | 14.9% | 22.2% |
| Vocational | | 10.5% | 11.2% |
| O level | | 36.9% | 33.8% |
| A level | | 24.0% | 20.7% |
| Degree | | 13.7% | 12.1% |
| Social Class | 4876 | | |
| I Professionals | | 5.9% | 5.8% |
| II Managerial and technical | | 31.4% | 25.4% |
| IIIN Skilled Non-manual | | 42.8% | 38.3% |
| IIIM Skilled Manual | | 7.9% | 8.0% |
| IV Partly skilled manual | | 9.8% | 12.0% |
| V Unskilled Manual or Armed Forces | | 2.2% | 10.4% |
| Parity | 2000 | 0.8 (1.0) | 0.9 (0.01) |
| Edinburgh Post-Natal Depression Score | 3810 | 5.4 (4.7) | 6.0 (0.1) |
| Partner's baseline alcohol use disorder | 4254 | | |
| No | | 98.9% | 91.9% |
| Yes | | 1.1% | 8.1% |

(RF) method in R. For this analysis, we ran five imputations. All continuous predictors were categorized using the cut-off values detailed in the technical appendix (List B in the S1 Appendix).

Our sensitivity analysis had 14,734 observations. One observation in the data was missing all study variables except the child's BMI–a sex-specific BMI category could not be computed so the observation was dropped. The data imputed from the random forest method were then compiled and analyzed in SAS with the same methods as the primary analysis.

A graphical comparison of the differences in prevalence across time of the four primary study variables between the original data with missing values and each imputed dataset (FCS

and RF) is available in the appendix (S2 Fig in the S1 Appendix). We further stratified the primary study variables by outcome (S3 Fig in the S1 Appendix).

## Statistical methods

Spaghetti plots were used to map the temporal patterning of IPV over the life course, and identify differences over time between ever homeless and never homeless mothers. Expected risks of homelessness under different patterns of emotional and physical IPV were estimated using multiple imputation to adjust for selection bias and censoring, and marginal structural survival models [30, 31] to adjust for time-varying confounding.

The prevalence of emotional, physical, and any IPV were plotted over each wave of data collection to identify life course trends. These trend lines were further stratified by the outcome measure of homelessness. Each wave of data captured responses that refer to a months-long time interval. For a given interval, the data do not tell us whether financial adversity occurred prior to or after emotional and physical IPV. For our primary analysis (Model 1), we assumed that financial adversity occurs before IPV in each time point, and included concurrent financial adversity as a predictor of emotional and physical IPV in the treatment weights. This assumption more conservatively estimates the effect of IPV by assuming that within-interval financial adversity attenuates its role. As a sensitivity analysis (Model 2), we re-ran our analysis with the assumption that IPV occurs before financial adversity in each time point, instead including concurrent emotional IPV and physical IPV as predictors of financial adversity in the treatment weights.

To estimate the parameters of the marginal structural model, we used inverse probability weighting to account for time-varying confounding between our three exposures: emotional IPV, physical IPV, and financial adversity. For the primary analysis, longitudinal participant histories in the data were converted to person-time format. Then, each person-row in the data was associated with a stabilized treatment weight, computed from the product of the probabilities of that participant's exposure experiences, from baseline through that row's associated time-point, conditional on time-varying history of exposure and covariates. We then fit a weighted pooled logistic regression model using a categorical indicator of time to estimate discrete-time hazards of incident homelessness, using proc genmod with the repeated statement to obtain robust standard errors.

We included two-way and three-way interaction terms between our exposure variables to assess whether experiencing the exposures jointly compounds or reduces the risk of subsequent homelessness. While Von Hippel [32] recommends transforming, then imputing, interaction terms, adding three interaction terms for each of eight waves of the study, given the sparsity and collinearity of the original multi-level data, greatly affected the joint distribution of the data. This led to unusual and implausible imputed values, so we imputed, then transformed our interaction terms.

We repeated the full analysis on the data imputed from MICE using RF in order to assess whether our estimates were robust to model specification in the imputation of missing data.

The estimates across imputations were compiled according to Rubin's rules in SAS using the proc mianalyze command, with the edf option set to 14,723 degrees of freedom for each parameter estimate.

## Results

### Sample characteristics

Table 1 presents descriptive statistics of the study sample, including the number of observations missing for each variable in the original data, the distributions of each variable in the original data, and the distributions of each variable in the primary data from MICE using FCS.

The study population was majority white, with an average age of 27.7 years at baseline. A plurality of individuals had completed education up to O-levels, and identified in the social class skilled non-manual labor. Average parity was 0.9, average EPDS score was 6.0, and approximately 8% of partners had baseline alcohol use disorder. Approximately 37% of women experienced emotional IPV and 20% physical IPV. Fifty-five percent had experienced any financial adversity over the course of the study, while almost 13% experienced any homelessness.

These imputed values differed from the original data, indicating that the women with missing values differed in their observable data from women without missing values. The difference consistently suggests that women with missing data were more likely to be socially and financially vulnerable, indicated by lower age, more non-white, lower education and social class, higher parity, higher EPDS score, and higher partner alcohol use disorder. Prevalence rates of IPV, financial adversity, and homelessness were all substantially higher in the women with missing data than in those with non-missing data.

Fig 1 contrasts the life course trends of women who were homeless during the study period and those who were not. The reported rates of exposure for homeless women are notably higher than for women who did not become homeless. Prevalence for both groups peaks at wave 5 (about 46% for ever homeless mothers, versus 14% for never homeless mothers). The most notable trend, unique to the group of homeless mothers, is a large dip in reported physical IPV in wave 3 (at a prevalence of 7%). This time period covers the 8 months following childbirth. This dip is not mirrored in the reported rates of emotional IPV.

### Regression estimates

Table 2 displays the results of our marginal structural models.

**Model 1.** Model 1 assumed that financial adversity occurs before IPV in each time interval. In Model 1, the estimated marginal effect of emotional IPV on incident homelessness over the study period was a 1.44 (95% CI: 1.13–1.84) times increase in the discrete-time hazard of homelessness for every additional experience of emotional IPV. The estimated effect of each report of physical IPV was a 2.05 (95% CI: 1.21–3.49) times increase in homelessness. The

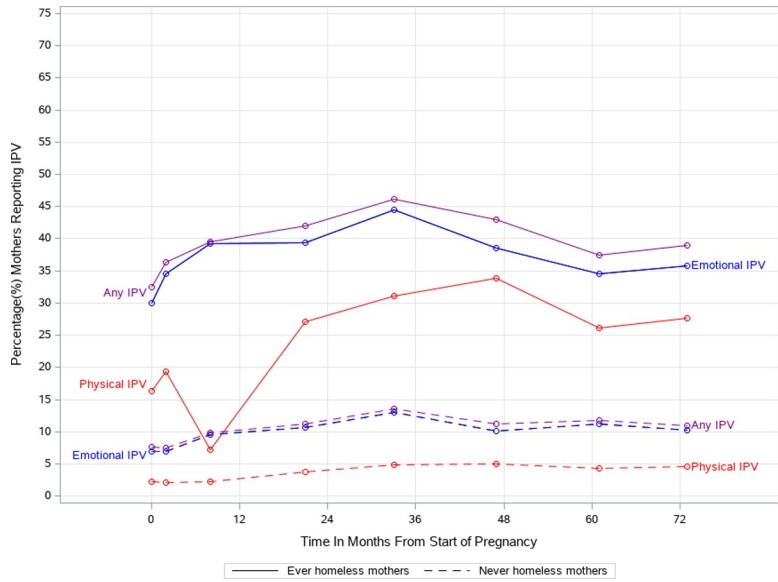

**Fig 1. Prevalence of IPV among mothers in 82 months following start of pregnancy, stratified by homeless status.**

**Table 2. Associations between emotional and physical IPV and financial adversity on homelessness.**

| | Hazard Ratio | 95% Confidence Interval | Sensitivity Analysis | |
|---|---|---|---|---|
| | | | MICE with RF | |
| | | | Hazard Ratio | 95% Confidence Interval |
| Model 1 (assuming financial adversity occurs before IPV in each interval) | | | | |
| A. Emotional IPV only | **1.44** | **(1.13,1.84)** | 1.77 | (1.51,2.08) |
| B. Physical IPV only | **2.05** | **(1.21,3.49)** | 4.12 | (2.98,5.68) |
| C. Financial adversity only | **1.59** | **(1.44,1.77)** | 1.81 | (1.67,1.95) |
| D. Emotional and physical IPV | 2.51 | (0.99,6.33) | **4.84** | **(2.49,9.42)** |
| E. Emotional IPV and financial adversity | **2.09** | **(1.35,3.22)** | 2.67 | (1.90,2.75) |
| F. Physical IPV and financial adversity | **2.79** | **(1.21,6.44)** | 5.55 | (2.93,10.53) |
| G. Emotional and physical IPV and financial adversity | 3.21 | (0.83,12.40) | 5.92 | (1.85,18.97) |
| Model 2 (assuming IPV occurs before financial adversity in each interval) | | | | |
| A. Emotional IPV only | **1.45** | **(1.17,1.81)** | 1.77 | (1.51,2.07) |
| B. Physical IPV only | **2.34** | **(1.47,3.72)** | 4.25 | (3.05,5.91) |
| C. Financial adversity only | **1.61** | **(1.46,1.77)** | 1.80 | (1.66,1.94) |
| D. Emotional and physical IPV | **2.81** | **(1.23,6.43)** | 5.05 | (2.62,9.72) |
| E. Emotional IPV and financial adversity | **2.13** | **(1.45,3.14)** | 2.69 | (1.94,3.73) |
| F. Physical IPV and financial adversity | **3.18** | **(1.53,6.63)** | 6.25 | (3.25,12.03) |
| G. Emotional and physical IPV and financial adversity | **3.63** | **(1.10,12.02)** | 6.66 | (2.12,20.91) |

\* All models are adjusted for socio-demographic variables age, ethnicity, education, partner's baseline alcohol use disorder, parity, postpartum depression, and social class.

estimated effect of each report of financial adversity was a 1.59 (95% CI: 1.44–1.77) times increase in homelessness.

The estimated marginal joint effect of emotional IPV and financial adversity on incident homelessness over the study period was a 2.09 (95% CI: 1.35–3.22) times multiplicative increase in the discrete-time hazard of homelessness for every additional experience of concurrent emotional IPV and financial adversity. The estimated effect of each report of concurrent physical IPV and financial adversity was a 2.79 (95% CI: 1.21–6.44) times increase in homelessness.

There did not appear to be an interaction effect between emotional and physical IPV in our analysis. We also did not see evidence of a three-way interaction between emotional IPV, physical IPV, and financial adversity.

The results of Model 2 are available in Part C of the S1 Appendix.

**Sensitivity analysis.** The summary statistics for the sensitivity analysis conducted on the RF imputed data are available in S2 Table in the S1 Appendix. The results of the analysis are in Part D of the S1 Appendix.

## Discussion

Our results provide evidence that emotional IPV, physical IPV, and financial adversity each increase risk of homelessness. Emotional IPV and financial adversity had comparable effects of roughly 1.5–1.8 times increased hazard rate of homelessness, while physical IPV had a larger effect on homelessness, between 2–4 times increased hazard of homelessness. We also observed interactions of emotional IPV with concurrent financial adversity, and physical IPV with concurrent financial adversity. The interactions increased the hazard rate of homelessness by roughly 2.1 and 2.8–3.2, respectively.

The design of our study gave us a unique perspective on life course trends by centering around a major life event, the birth of a child, rather than on chronological age. This led to a notable finding, unique to mothers who became homeless, that physical IPV dropped drastically and temporarily in the period from childbirth through the first eight months of the child's life.

There is mixed existing evidence supporting this trend. The 1 year and 3 month postpartum periods have been linked with increased IPV from pregnancy [33–35]. IPV may emerge in the second half, rather than the first half, of the postpartum year [36]. However, other studies align with our finding of a decrease in IPV both during pregnancy [37] and in the postpartum period [38]. From a policy perspective, instituting screening for IPV in the postpartum year may miss mothers who are high-risk.

Our results align with existing literature that estimates 1.14 to 4 times increased odds of housing instability or homelessness [18–20] associated with IPV. We appear to be the first to separate IPV into emotional and physical components, and to examine the contribution of concurrent financial adversity in modifying the effect on homelessness. Given that financial adversity is an obvious and direct cause of homelessness, our results highlight the severity of the effect of IPV in victims' lives. These effects are compounded when the victim experiences financial hardship, and lead to severe vulnerability to becoming homeless.

Our findings were limited by a number of factors. Recall and non-response bias likely affected surveys mailed to home addresses, particularly for participants sharing a residence with an abusive partner, or for homeless women. Furthermore, each wave of data collection measured a different interval of time; our reporting scale could underweight data from longer reporting periods.

Although we used robust statistical methods to address the limitations of our data, large differences emerged between our two imputation methods. The FCS imputed data estimated greater prevalence of IPV and financial adversity at each time point and across the study. The RF imputed data estimated greater overall prevalence of IPV and financial adversity across the study, but lower prevalence within each time interval. The FCS method is susceptible to model misspecification–it is possible that the predictor variables we chose overestimated the association between our primary study variables and non-response. The RF method, based on decision trees, may "regress towards the mean" within classification groups, which could inflate estimates in the unbalanced data in this study. This could explain the much larger effect sizes in the RF imputed data for our relatively rare exposure and outcome variables, compared to the FCS imputed data.

The difficulties we experienced in producing stable estimates of missing data align with our understanding that both IPV and homelessness are undercounted and susceptible to measurement error. Our study reinforces the need for research on these topics to account for biases with robust statistical methods.

Despite the differences in our analyses between the imputed datasets, our results are congruent in identifying clear and nuanced effects of emotional IPV and physical IPV on homelessness, interactions with financial adversity, and important patterns of exposure over time.

Our findings may serve as a tool to sharpen policies intended to prevent familial homelessness. In the current COVID-19 pandemic, widespread economic hardship may be contributing to a rise in IPV during this time [39, 40], and underscores the importance of housing policies that allow victims to separate from abusers. The landscape of policies and funding in the US to provide screening and services to support at-risk women remains piecemeal. The Violence Against Women Act, which includes rules protecting domestic violence victims in Housing and Urban Development programs, expired, was briefly reinstated, and expired again in 2018 and 2019 [41]. Funding to reduce homelessness through the McKinney-Vento Act is reactive rather than preventive.

More broadly, this study elucidates the value of approaching social and macroeconomic policies on homelessness with detailed evidence for the specific realities of vulnerable sub-groups to best guide effective decision-making. There is not a one-size-fits-all approach. We provide a novel understanding of the intersectional reality of pregnancy and risks of IPV victimization and familial homelessness.

We believe our thorough statistical approach to addressing the limitations of data is an underutilized and essential avenue of research to combat the pervasive effects of violence on the lives of vulnerable women.

## Supporting information

**S1 Appendix.**
(DOCX)

## Acknowledgments

We are extremely grateful to all the families who took part in this study, the midwives for their help in recruiting them, and the whole ALSPAC team, which includes interviewers, computer and laboratory technicians, clerical workers, research scientists, volunteers, managers, receptionists and nurses.

## Author Contributions

**Conceptualization:** Caitlin S. Chan.

**Data curation:** Caitlin S. Chan.

**Formal analysis:** Caitlin S. Chan, Aaron L. Sarvet.

**Funding acquisition:** Aaron L. Sarvet, Archana Basu, Karestan Koenen.

**Investigation:** Caitlin S. Chan.

**Methodology:** Caitlin S. Chan, Aaron L. Sarvet.

**Project administration:** Caitlin S. Chan, Archana Basu, Karestan Koenen.

**Resources:** Archana Basu, Karestan Koenen.

**Software:** Caitlin S. Chan, Aaron L. Sarvet.

**Supervision:** Archana Basu, Karestan Koenen, Katherine M. Keyes.

**Validation:** Caitlin S. Chan.

**Visualization:** Caitlin S. Chan.

**Writing – original draft:** Caitlin S. Chan.

**Writing – review & editing:** Caitlin S. Chan, Aaron L. Sarvet, Archana Basu, Karestan Koenen, Katherine M. Keyes.

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
