## [Decision Letter · Decision Letter 0]

25 Nov 2020

PONE-D-20-34305

Associations of intimate partner violence and financial adversity with familial homelessness in pregnant and postpartum women: A 7-year prospective study of the ALSPAC cohort

PLOS ONE

Dear Dr. Chan,

Thank you for submitting your manuscript to PLOS ONE. After careful consideration, we feel that it has merit but does not fully meet PLOS ONE’s publication criteria as it currently stands. Therefore, we invite you to submit a revised version of the manuscript that addresses the points raised during the review process.

We look forward to receiving your revised manuscript.

Kind regards,

Marianna Mazza

Academic Editor

PLOS ONE

Journal Requirements:

"Ethical approval for the study was obtained from the ALSPAC Ethics and Law Committee and the Local Research Ethics Committees (Reference number C2481), and all participants provided written informed consent."   

For additional information about PLOS ONE ethical requirements for human subjects research, please refer to " ext-link-type="uri" xlink:type="simple">http://journals.plos.org/plosone/s/submission-guidelines#loc-human-subjects-research."

3. In your Methods section, please provide additional information about the participant recruitment method and the demographic details of your participants. Please ensure you have provided sufficient details to replicate the analyses such as:

a) the recruitment date range (month and year),

b) a description of how participants were recruited, and c) descriptions of where participants were recruited and where the research took place.

4. Please include additional information regarding the survey or questionnaire used in the study and ensure that you have provided sufficient details that others could replicate the analyses. For instance, if you developed a questionnaire as part of this study and it is not under a copyright more restrictive than CC-BY, please include a copy, in both the original language and English, as Supporting Information.

Reviewers' comments:

Reviewer's Responses to Questions

**Comments to the Author**

1. Is the manuscript technically sound, and do the data support the conclusions?

Reviewer #1: Yes

2. Has the statistical analysis been performed appropriately and rigorously? 

Reviewer #1: Yes

3. Have the authors made all data underlying the findings in their manuscript fully available?

Reviewer #1: Yes

4. Is the manuscript presented in an intelligible fashion and written in standard English?

Reviewer #1: Yes

5. Review Comments to the Author

Reviewer #1: The manuscript describes the relationship of intimate partner violence and financial adversity with familial homelessness in pregnant and postpartum women.

The theme is adequately explored and with a flowing form. It could be stimulating a reflection on the victimization process and the connected psychopathological configurations, as well as on the fact that challenging circumstances and resilience skills can contribute to a better coping with stress and adversity.

It might be interesting to analyze the impact of COVID19 pandemic on emotional distress and sexuality. It would therefore be useful to read and discuss the following papers:

G Marano, E Gaetani, A Gasbarrini, L Janiri, G Sani, M Mazza, Multidisciplinary Gemelli Group for HHT. Mental health and counseling intervention for hereditary hemorrhagic telangiectasia (HHT) during the COVID-19 pandemic: perspectives from Italy. Eur Rev Med Pharmacol Sci. 2020 Oct;24(19):10225-10227. doi: 10.26355/eurrev_202010_23246.

M Mazza, G Marano, L Janiri, G Sani. Managing Bipolar Disorder patients during COVID-19 outbreak. Bipolar Disord. 2020 Oct 4. doi: 10.1111/bdi.13015.

M Mazza, G Marano, B Antonazzo, E Cavarretta, M Di Nicola, L Janiri, G Sani, G Frati, E Romagnoli. What about heart and mind in the covid-19 era? Minerva Cardioangiol. 2020 May 12. doi: 10.23736/S0026-4725.20.05309-8.

M Mazza, G Marano, C Lai, L Janiri, G Sani. Danger in danger: Interpersonal violence during COVID-19 quarantine. Psychiatry Res. 2020 Jul;289:113046. doi: 10.1016/j.psychres.2020.113046.

6. PLOS authors have the option to publish the peer review history of their article (what does this mean?). If published, this will include your full peer review and any attached files.

Reviewer #1: No

---

## [Author Response · Author response to Decision Letter 0]

16 Dec 2020

December 8, 2020

Marianna Mazza

Academic Editor

PLOS ONE

Dear Dr. Marianna Mazza,

We would like to thank you for your response letter and the opportunity to resubmit a revised copy of this manuscript. We express our thanks for the positive feedback and helpful comments for correction and modification. We believe the modifications have resulted in an improved revised manuscript, which you will find uploaded alongside this document. The manuscript has been revised to address the reviewer comments. We state below the steps we have taken to address the points raised in your letter.

We very much hope the revised manuscript is accepted for publication in PLOS ONE.

Sincerely,

Caitlin Chan, on behalf of the authors

Journal Requirements:

 We corrected the formatting on the title page. We modified the spacing in the tables, the formatting of the table and figure citations, and the order of the paper sections.

"Ethical approval for the study was obtained from the ALSPAC Ethics and Law Committee and the Local Research Ethics Committees (Reference number C2481), and all participants provided written informed consent." 

For additional information about PLOS ONE ethical requirements for human subjects research, please refer to http://journals.plos.org/plosone/s/submission-guidelines#loc-human-subjects-research."

We have revised the ethics statement as follows:

“Ethical approval for the study was obtained from the ALSPAC Ethics and Law Committee and the Local Research Ethics Committees (Reference number C2481), and all participants provided written informed consent. ALSPAC Executive committee has been registered with/has an Institutional Review Board status with the United States Department of Health and Human Services and Office for Human Research Protections (IRB I00003312). The ALSPAC committee approval serves as the IRB approval.”

This statement has been updated in both the methods section and the ethics statement of the submission file.

3. In your Methods section, please provide additional information about the participant recruitment method and the demographic details of your participants. Please ensure you have provided sufficient details to replicate the analyses such as:

a) the recruitment date range (month and year),

b) a description of how participants were recruited, and c) descriptions of where participants were recruited and where the research took place.

 We added the recruitment location, date range, and additional description of the recruitment process of the original cohort. We further moved the study exclusion criteria from the appendix to the methods section of the paper.

4. Please include additional information regarding the survey or questionnaire used in the study and ensure that you have provided sufficient details that others could replicate the analyses. For instance, if you developed a questionnaire as part of this study and it is not under a copyright more restrictive than CC-BY, please include a copy, in both the original language and English, as Supporting Information.

 We have included in the methods section a link to the questionnaires.

Access to ALSPAC research data must be requested using the formal procedures described in this document (http://www.bristol.ac.uk/media-library/sites/alspac/documents/researchers/data-access/ALSPAC_Access_Policy.pdf) and is subject to eligibility, the ALSPAC funder’s terms and conditions and University of Bristol policies and procedures. Data are available upon request from the ALSPAC executive committee (alspac-exec@bristol.ac.uk) for researchers who meet the criteria for access.

Reviewers' comments:

Reviewer's Responses to Questions

Comments to the Author

1. Is the manuscript technically sound, and do the data support the conclusions?

Reviewer #1: Yes

2. Has the statistical analysis been performed appropriately and rigorously?

Reviewer #1: Yes

3. Have the authors made all data underlying the findings in their manuscript fully available?

Reviewer #1: Yes

4. Is the manuscript presented in an intelligible fashion and written in standard English?

Reviewer #1: Yes

5. Review Comments to the Author

Reviewer #1: The manuscript describes the relationship of intimate partner violence and financial adversity with familial homelessness in pregnant and postpartum women.

The theme is adequately explored and with a flowing form. It could be stimulating a reflection on the victimization process and the connected psychopathological configurations, as well as on the fact that challenging circumstances and resilience skills can contribute to a better coping with stress and adversity.

It might be interesting to analyze the impact of COVID19 pandemic on emotional distress and sexuality. It would therefore be useful to read and discuss the following papers:

G Marano, E Gaetani, A Gasbarrini, L Janiri, G Sani, M Mazza, Multidisciplinary Gemelli Group for HHT. Mental health and counseling intervention for hereditary hemorrhagic telangiectasia (HHT) during the COVID-19 pandemic: perspectives from Italy. Eur Rev Med Pharmacol Sci. 2020 Oct;24(19):10225-10227. doi: 10.26355/eurrev_202010_23246.

M Mazza, G Marano, L Janiri, G Sani. Managing Bipolar Disorder patients during COVID-19 outbreak. Bipolar Disord. 2020 Oct 4. doi: 10.1111/bdi.13015.

M Mazza, G Marano, B Antonazzo, E Cavarretta, M Di Nicola, L Janiri, G Sani, G Frati, E Romagnoli. What about heart and mind in the covid-19 era? Minerva Cardioangiol. 2020 May 12. doi: 10.23736/S0026-4725.20.05309-8.

M Mazza, G Marano, C Lai, L Janiri, G Sani. Danger in danger: Interpersonal violence during COVID-19 quarantine. Psychiatry Res. 2020 Jul;289:113046. doi: 10.1016/j.psychres.2020.113046.

We cited the last paper, “Danger in danger: Interpersonal violence during COVID-19 quarantine,” which discusses a potential rise in interpersonal violence that may exacerbate the conditions discussed in our study. The other studies are not directly relevant to the subject matter of our paper.

6. PLOS authors have the option to publish the peer review history of their article (what does this mean?). If published, this will include your full peer review and any attached files.

Do you want your identity to be public for this peer review? For information about this choice, including consent withdrawal, please see our Privacy Policy.

Reviewer #1: No

---

## [Decision Letter · Decision Letter 1]

2 Jan 2021

Associations of intimate partner violence and financial adversity with familial homelessness in pregnant and postpartum women: A 7-year prospective study of the ALSPAC cohort

PONE-D-20-34305R1

Dear Dr. Chan,

We’re pleased to inform you that your manuscript has been judged scientifically suitable for publication and will be formally accepted for publication once it meets all outstanding technical requirements.

Kind regards,

Marianna Mazza

Academic Editor

PLOS ONE

Additional Editor Comments (optional):

Reviewers' comments:

Reviewer's Responses to Questions

**Comments to the Author**

1. If the authors have adequately addressed your comments raised in a previous round of review and you feel that this manuscript is now acceptable for publication, you may indicate that here to bypass the “Comments to the Author” section, enter your conflict of interest statement in the “Confidential to Editor” section, and submit your "Accept" recommendation.

Reviewer #2: All comments have been addressed

2. Is the manuscript technically sound, and do the data support the conclusions?

Reviewer #2: (No Response)

3. Has the statistical analysis been performed appropriately and rigorously? 

Reviewer #2: (No Response)

4. Have the authors made all data underlying the findings in their manuscript fully available?

Reviewer #2: (No Response)

5. Is the manuscript presented in an intelligible fashion and written in standard English?

Reviewer #2: (No Response)

6. Review Comments to the Author

Reviewer #2: (No Response)

7. PLOS authors have the option to publish the peer review history of their article (what does this mean?). If published, this will include your full peer review and any attached files.

Reviewer #2: No

---

## [Editor Report · Acceptance letter]

6 Jan 2021

PONE-D-20-34305R1 

Associations of intimate partner violence and financial adversity with familial homelessness in pregnant and postpartum women: A 7-year prospective study of the ALSPAC cohort 

Dear Dr. Chan:

I'm pleased to inform you that your manuscript has been deemed suitable for publication in PLOS ONE. Congratulations! Your manuscript is now with our production department. 

Kind regards, 

on behalf of

Dr. Marianna Mazza 

Academic Editor

PLOS ONE